# C-CLIP: Multimodal Continual Learning for Vision-Language Model

**Wenzhuo Liu**[1,2*]   **Fei Zhu**[3†]   **Longhui Wei**[4†]   **Qi Tian**[4]

[1]School of Artificial Intelligence, UCAS
[2]State Key Laboratory of Multimodal Artificial Intelligence Systems, CASIA
[3]Centre for Artificial Intelligence and Robotics, HKISI-CAS    [4]Huawei Inc
{liuwenzhuo2020, zhufei2018}@ia.ac.cn, weilh2568@gmail.com

## Abstract

Multimodal pre-trained models like CLIP need large image-text pairs for training but often struggle with domain-specific tasks. Since retraining with specialized and historical data incurs significant memory and time costs, it is important to continually learn new domains in the open world while preserving original performance. However, current continual learning research mainly focuses on unimodal scenarios, and the evaluation criteria are insufficient without considering image-text matching performance and the forgetting of zero-shot performance. This work introduces image-caption datasets from various domains and establishes a multimodal vision-language continual learning benchmark. Then, a novel framework named C-CLIP is proposed, which not only prevents forgetting but also enhances new task learning impressively. Comprehensive experiments demonstrate that our method has strong continual learning ability across diverse image-text datasets, maintaining zero-shot prediction capabilities with minimal forgetting and significantly outperforming existing methods.

## 1 Introduction

Multimodal pre-trained models like CLIP (Radford et al., 2021) have recently gained widespread attention for providing general visual-language representations on downstream tasks such as image question answering, classification, semantic segmentation, and object detection. Although CLIP is trained on a large number of image-text pairs, it still struggles to handle image-text pairs from unseen domains (Zhang et al., 2024), limiting real-world applicability. A straightforward way is to fine-tune the pre-trained CLIP model on the domain-specific dataset. However, this often leads to catastrophic forgetting (French, 1999) of existing knowledge, including both the CLIP's original general representations (i.e., zero-shot generalization) and the knowledge on other learned tasks. Besides, the high memory and training cost makes retraining CLIP infeasible. Therefore, a natural, fundamental yet underexplored question is *how to maintain general representations of visual-language model while adapting to new domains continually?*

Continual learning (CL) (Liu et al., 2025; Hou et al., 2019; Yan et al., 2021; Gao et al., 2023) has explored the problem of retaining old knowledge while learning new tasks. However, current CL studies have several key challenges: **Firstly**, compared to standard unimodal continual learning scenarios, multimodal settings present greater complexity. Although CLIP can handle both image classification and purely multimodal tasks, the challenges of addressing forgetting and adaptability differ significantly between these two tasks. For image classification, pre-trained models often exhibit strong zero-shot performance. As a result, many previous approaches Wang et al. (2022); Tang et al. (2023); D'Alessandro et al. (2023); Qiao et al. (2023); Li et al. (2024); Wang et al. (2024) rely on prompt-based designs, keeping parameters fixed without fine-tuning the model itself. In contrast, multimodal tasks are far more challenging, as poor performance in a specific domain often necessitates simultaneous fine-tuning of both the vision and text encoders.

---

*This work was done during an internship at Huawei Inc.

†Corresponding author. Code available at https://github.com/SmallPigPeppa/C-CLIP

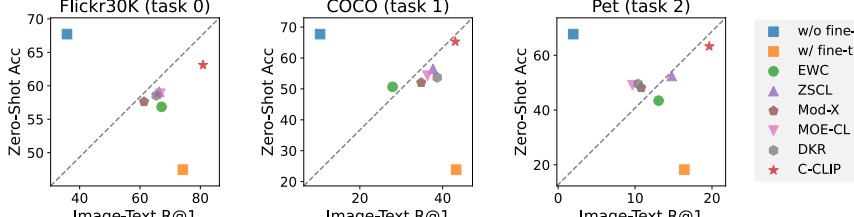

Figure 1: Performance comparison on downstream tasks and the general representation ability. The pre-trained CLIP (Radford et al., 2021) has good ImageNet zero-shot accuracy but performs poorly on downstream tasks. Directly full fine-tuning remarkably enhances the image-text retrieval performance on downstream tasks, while leading to catastrophic forgetting of ImageNet zero-shot ability. Existing methods generally seek a trade-off between zero-shot generalization and downstream task performance. Differently, our method (C-CLIP) impressively achieves strong downstream task performance (even outperforms full fine-tuning) and well preserves the general representation ability.

**Secondly**, the evaluation of vision-language models (VLMs) remains insufficient. While some recent works (Zheng et al., 2023; Yu et al., 2024; Srinivasan et al., 2022) have leveraged pre-trained vision-language models like CLIP in tasks such as few-shot class-incremental learning D'Alessandro et al. (2023), visual question-answering Qian et al. (2023), or cross-modal retrieval Wang et al. (2021), their evaluations often focus on specific aspects, such as image classification performance or cross-modal retrieval accuracy. We aim to provide a more comprehensive evaluation by assessing both the general zero-shot capabilities and their continual learning performance in downstream multimodal tasks.

**Third**, traditional CL methods (Kirkpatrick et al., 2017; Zenke et al., 2017; Li & Hoiem, 2017; Hou et al., 2019; Douillard et al., 2020) apply regularization to reduce forgetting, which unfortunately hinders the learning of new tasks. In other words, these methods forget less because they learn less, losing the plasticity in the continual learning process gradually (Dohare et al., 2024). Therefore, we seek to leverage the properties of multimodal representation learning to enable simultaneous learning of new and old knowledge, overcoming past trade-offs.

In this work, we establish a Vision-Language continual learning (VLCL) benchmark based on image-text datasets (e.g., Flickr30K (Plummer et al., 2015) and COCO-caption (Chen et al., 2015)), and propose the evaluation on three aspects: image-text retrieval on downstream tasks, retrieval in unseen domains, and the general performance of visual-language model. Then, we propose a multimodal continual learning approach named C-CLIP. Specifically, we demonstrate that reducing trainable parameters can yield similar results to the existing sophisticated CL method, and simplify the previously complex strategies with low-rank adaption (LoRA) (Hu et al.). In addition, we experimentally find that existing methods (Kirkpatrick et al., 2017; Li & Hoiem, 2017; Douillard et al., 2020; Zheng et al., 2023; Yu et al., 2024) generally seek a trade-off between zero-shot generalization and downstream task performance. To overcome this limitation, we propose contrastive knowledge consolidation that not only reduces forgetting of old tasks but also enhances learning on new tasks, even matching or exceeding full fine-tuning performance. The results in Figure 1 show that our method significantly improves the performance on downstream tasks while generally preserving the ImageNet zero-shot accuracy.

Our main contributions are as follows:

- We introduce a benchmark for continual multimodal representation learning, emphasizing that visual-language model models should retain their original general performance and learn new image-text data simultaneously.

- We propose the C-CLIP method that consists of multimodal low-rank adaptation and contrastive knowledge consolidation, achieving the goal of learning more and forgetting less for the first time.

- Extensive experiments demonstrate that our method significantly enhances performance across different domain image-text datasets without catastrophic forgetting.

## 2 RELATED WORK

### 2.1 CONTINUAL LEARNING

Continual Learning (Kirkpatrick et al., 2017), also known as incremental learning, has received extensive attention recently. Existing research mainly focuses on unimodal tasks like supervised image classification, with class-incremental learning (CIL) (Masana et al., 2022) as the common benchmark. Regularization methods (Kirkpatrick et al., 2017; Zenke et al., 2017; Li & Hoiem, 2017) mainly constrain the changes of parameters or feature spaces. Data replay methods (Rebuffi et al., 2017; Hou et al., 2019) store and replay a subset of old data, which leads to additional computational, memory, and privacy costs. Architecture-based methods (Schwarz et al., 2018; Yan et al., 2021) add new models for each task, but their parameters grow fast with more tasks, making them unsuitable for real-world applications. More recent rehearsal-free approaches (Wang et al., 2022; Tang et al., 2023; D'Alessandro et al., 2023; Guo et al., 2024; Li et al., 2024; Wang et al., 2024) explore parameter-efficient strategies (e.g., prompt tuning, prefix tuning) for continual fine-tuning of pre-trained models. However, the above methods mainly focus on unimodal scenarios, ignoring widely existing multimodal tasks (Ramachandram & Taylor, 2017) in real-world applications.

A few recent studies (Zheng et al., 2023; Yu et al., 2024) involve visual-language models in continual learning. Specifically, Jin et al. (2020) studied the visually-grounded masked language prediction task by learning from streaming visual scenes. Zheng et al. (2023) and Yu et al. (2024) studied the multi-domain task incremental learning (MTIL) benchmark that continually fine-tunes CLIP on visual datasets and evaluates the zero-shot performance of new tasks. However, it is still far from the goal of continually updating multimodal pre-trained models. Mod-X (Ni et al., 2023) is most relevant to our work, which performs stage-wise data continual training on a fixed image-text dataset. However, each task shares the same data distribution, which differs from our goal of adapting CLIP to diverse domains. Additionally, none of these works considers CLIP's original (e.g., zero-shot generalization) performance during continual learning. In this paper, we study a more realistic and challenging scenario named multimodal continual learning with the visual-language model (VLCL), and a comparison of these settings is shown in Table 1.

Table 1: Comparison of CIL, MTIL, and VLCL benchmark.

| Setting | Train Data | Eval Metric | Eval Origin Performance? |
|---------|-----------|-------------|--------------------------|
| CIL | Image, Label | Classification Acc | ✗ |
| MTIL | Image, Label | Zero-shot Acc (only new tasks) | ✗ |
| VLCL (ours) | Image, Caption | Image-Text Recall & Zero-shot Acc | ✓ |

### 2.2 VISUAL-LANGUAGE REPRESENTATIONAL LEARNING

Vision-language representation learning (Radford et al., 2021; Jia et al., 2021; Zhang et al., 2024) has gained widespread attention across various fields. Among them, CLIP (contrastive language-image pretraining) (Radford et al., 2021) performs exceptionally well across many downstream tasks such as VQA (Antol et al., 2015), classification (He et al., 2016), and detection (Ren et al., 2016). CLIP consists of an image encoder and a text encoder. During the pre-training stage, paired image-text samples are treated as positives, while unpaired samples are used as negatives under the contrastive learning paradigm (Chen et al., 2020). Despite the impressive performance, training CLIP relies on large datasets like Laion-400M (Schuhmann et al., 2021) and Conceptual Captions (Sharma et al., 2018), making it resource-intensive. In addition, although these large pre-training datasets cover diverse samples, well-trained vision-language models still struggle to match domain-specific image-text pairs (Zhang et al., 2024). Therefore, continually fine-tuning CLIP without losing its original performance becomes a key issue in real-world scenarios.

## 3 PROBLEM DEFINITION AND BENCHMARK

**Notation and problem definition.** Vision-Language continual learning (VLCL) involves learning from a sequence of $T$ tasks. At each stage $t \in \{1, ..., T\}$, the model is fine-tuned on an image-caption dataset $\mathcal{D}^t = \{(\boldsymbol{v}_i^t, \boldsymbol{c}_i^t)\}_{i=1}^{n_t}$, where $(\boldsymbol{v}_i^t, \boldsymbol{c}_i^t)$ represents an image-caption pair and $n_t$ is the

number of pairs in this dataset. In this paper, we focus on vision-language model like CLIP and represent model into two components: **(1)** a vision encoder $f_{\boldsymbol{\theta}} : \mathcal{V} \to \mathcal{Z}$, parameterized by $\boldsymbol{\theta}$, that transforms an image $\boldsymbol{v}$ into a feature vector $\boldsymbol{z}_v = f_{\boldsymbol{\theta}}(\boldsymbol{v})$ in a high-dimensional space $\mathcal{Z} \subset \mathbb{R}^d$. **(2)** a text encoder $g_{\boldsymbol{\varphi}} : \mathcal{C} \to \mathcal{Z}$, with parameters $\boldsymbol{\varphi}$, which maps a caption $\boldsymbol{c}$ to a feature vector $\boldsymbol{z}_c = g_{\boldsymbol{\varphi}}(\boldsymbol{c})$.

VLCL aims to continually learn a function $f \circ g : \mathcal{V} \times \mathcal{C} \to \mathcal{Y}$ that can assign the correct label to each image-caption pair from all seen tasks. Specifically, at stage $t$, the challenge is to minimize a loss function $\ell$ (e.g., symmetric cross-entropy) on the new dataset $\mathcal{D}^t$, while preserving knowledge from previous tasks and potentially improving on earlier learning (Aljundi, 2019) as follows:

$$
\min_{\theta,\varphi,\epsilon} \mathbb{E}_{(\boldsymbol{v},\boldsymbol{c}) \backsim \mathcal{D}^t}[\ell(f_\theta(\boldsymbol{v}), g_\varphi(\boldsymbol{c}))] + \sum \epsilon_j
$$
$$
\text{s.t. } \mathbb{E}_{((\boldsymbol{v},\boldsymbol{c}) \backsim \mathcal{D}^j}[\ell(f_\theta(\boldsymbol{v}), g_\varphi(\boldsymbol{c})) - \ell(f_{\theta^{t-1}}(\boldsymbol{v}), g_{\varphi^{t-1}}(\boldsymbol{c}))] \leqslant \epsilon_j; \forall j \in \{1, 2, ..., t-1\}.
$$
(1)

The last term $\epsilon = \{\epsilon_j\}$ is a slack variable that can represent the forgetting ($\epsilon_j > 0$) or backward knowledge transfer ($\epsilon_j \leqslant 0$) on datasets $\mathcal{D}^j$ of $j$-th old task.

**VLCL benchmark.** To evaluate the continual learning performance of vision-language models, we establish a novel benchmark that includes three evaluation tracks, summarized in Table 2. **(1)** Multimodal continual learning. Eight image-caption datasets are used in this track. Among them, Flickr30K (Plummer et al., 2015) and COCO (Chen et al., 2015) are general real-world datasets. Other datasets, including Pets (Parkhi et al., 2012), Lexica (Shen et al., 2024), Simpsons, WikiArt (Saleh & Elgammal, 2015), Kream, and Sketch (Chowdhury et al., 2022), represent specific domains such as AI-generated images, art, clothing, and sketches. **(2)** Zero-shot retrieval. One held image-caption dataset, i.e., HAVG (Abdulmumin et al., 2022), is used to assess retrieval performance on unseen datasets. **(3)** Zero-shot classification. Previous CL work on CLIP has overlooked an important aspect: the forgetting of general representations (i.e., zero-shot generalization). To evaluate this, we tested zero-shot performance on six image classification datasets: ImageNet (Deng et al., 2009), CIFAR-100 (Krizhevsky et al., 2009), StanfordCars (Krause et al., 2013), Flowers (Nilsback & Zisserman, 2008), DTD (Cimpoi et al., 2014), and Food101 (Bossard et al., 2014).

Table 2: Evaluation setup of the VLCL benchmark datasets.

| Evaluation Aspect | Dataset |
|---|---|
| Multimodal Continual Learning | Flickr30K, COCO, Pets, Lexica, Simpsons, WikiArt, Kream, Sketch |
| Zero-shot Retrieval | HAVG |
| Zero-shot Classification | CIFAR-100, ImageNet, Flowers, DTD, Food101, StanfordCars |

**Evaluation metric.** We focus on multimodal continual learning of VLMs and view each dataset as one task, similar to the domain-incremental learning in classical CL (Van de Ven et al., 2022). The task identity (task-ID) is not needed at inference time. **(1)** For multimodal continual learning, we evaluate each dataset after fine-tuning all eight datasets (similar to the last accuracy in CIL), and report Recall@1 for both Image-to-Text (I2T R@1) and Text-to-Image (T2I R@1). The average performance on all datasets is also reported. **(2)** For zero-shot retrieval, we report I2T R@1. **(3)** For zero-shot classification, we report the performance during each stage of the continual learning process, as well as the final performance degradation (PD), which is calculated as the difference in accuracies between the original pre-trained CLIP model and the final fine-tuned model.

## 4 THE PROPOSED METHOD: C-CLIP

The core of continual learning is to preserve the performance of previous tasks and enhance the performance of new tasks. (1) For the first purpose, i.e., avoiding forgetting old tasks, without storing or replaying old data, existing studies (Kirkpatrick et al., 2017; Li & Hoiem, 2017; Zheng et al., 2023; Yu et al., 2024) have designed a variety of CL strategies, and the general idea is to use the current data to constrain the input-output relationship in old tasks. We empirically and theoretically show that low-rank adaptation could achieve a similar effect by reducing the number of trainable parameters. (2) For the second purpose, i.e., enhancing learning new tasks, we find that existing methods largely limit the plasticity of the model during continual learning, and we therefore

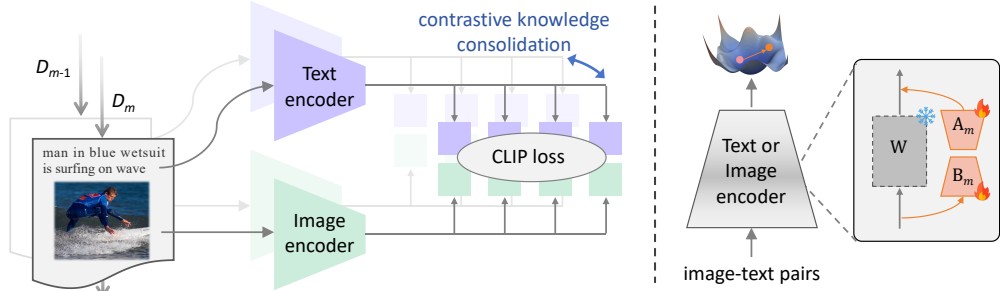

Figure 2: Illustration of the C-CLIP model. We make two adjustments to CLIP: **(1)** applying LoRA to reduce forgetting, though it slightly hampers new task learning; **(2)** introducing contrastive knowledge consolidation (CKC) improves learning on new tasks and reduces forgetting. This combination significantly enhances the continual learning ability of CLIP, achieving downstream task performance that matches or exceeds full fine-tuning and preserving its general zero-shot ability.

propose contrastive knowledge consolidation, which is designed for vision-language models. Our method is illustrated in Figure 2 and presented below in detail.

## 4.1 LoRA INTEGRATION FOR FORGETTING MITIGATION

**LoRA integration for VLCL.** To preserve the knowledge of previous tasks, we propose simply reducing the number of trainable parameters by leveraging parameter efficient tuning technique, i.e., low-rank adaptation (LoRA) (Hu et al.). As illustrated in Figure 2, LoRA composes of two rank decomposition matrices $\mathbf{B} \in \mathbb{R}^{u \times r}$ and $\mathbf{A} \in \mathbb{R}^{r \times v}$ where $r \in \mathbb{N}$ is the rank and $r \ll \min(u, v)$. $v$ and $u$ are the dimensionality of the input $\hat{\mathbf{x}} \in \mathbb{R}^v$ for current layer and hidden features, respectively. In this work, we apply LoRA in both vision encoder $f_\theta$ and text encoder $g_\varphi$ of CLIP. At each continual stage $t$, the previous parameters $\{\theta^{t-1}, \varphi^{t-1}\}$ remain frozen, while only the newly added $\mathbf{A}$ and $\mathbf{B}$ are trainable. However, since the model has no access to the task identity at inference, it is hard to decide which set of LoRA to use. More importantly, storing all the LoRAs for previous tasks leads to increasing memory issues. Therefore, we propose to integrate the current $\{\theta_{LoRA}, \varphi_{LoRA}\}$ into the main backbone at the end of each continual stage as follows:

$$\{\theta^t, \varphi^t\} = \{\theta^{t-1} + \alpha \cdot \theta_{LoRA}, \varphi^{t-1} + \alpha \cdot \varphi_{LoRA}\}, \tag{2}$$

where $\alpha \in [0, 1]$ is a pre-defined coefficient to reduce the forgetting of knowledge after integration.

**Theoretical analysis.** In the $t$-th continual stages, only the current dataset $\mathcal{D}^t$ is available, and an intuitive and general way to maintain the previous knowledge is to regulate the model by mimicking the input-output relation on previous $(t - 1)$-th continual stages based on $\mathcal{D}^t$ as follows:

$$\min_{\theta, \varphi} \mathbb{E}_{(\boldsymbol{v}, \boldsymbol{c}) \backsim \mathcal{D}^t}[\ell(f_\theta(\boldsymbol{v}), g_\varphi(\boldsymbol{c}))]$$
$$\text{s.t. } \mathbb{E}_{(\boldsymbol{v}, \boldsymbol{c}) \backsim \mathcal{D}^t}||f_\theta(\boldsymbol{v}) - f_{\theta^{t-1}}(\boldsymbol{v})|| \leqslant \epsilon, \quad \mathbb{E}_{(\boldsymbol{v}, \boldsymbol{c}) \backsim \mathcal{D}^t}||g_\varphi(\boldsymbol{c}) - g_{\varphi^{t-1}}(\boldsymbol{c})|| \leqslant \epsilon, \epsilon \geqslant 0. \tag{3}$$

For an arbitrary N-layer MLP model, assume that the activation function is bounded and Lipschitz continuous, and the input and all weights have bounded norms, then the model is Lipschitz continuous with respect to the weights (Appendix A.1 provides detailed proof), i.e., $||f_\theta(\boldsymbol{v}) - f_{\theta^{t-1}}(\boldsymbol{v})|| \leqslant K_f||\theta - \theta^{t-1}||$, then Eq. (3) can be rewritten as:

$$\min_{\theta, \varphi} \mathbb{E}_{(\boldsymbol{v}, \boldsymbol{c}) \backsim \mathcal{D}^t}[\ell(f_\theta(\boldsymbol{v}), g_\varphi(\boldsymbol{c}))]$$
$$\text{s.t. } ||\theta - \theta^{t-1}|| \leqslant \epsilon/K_f, \quad ||\varphi - \varphi^{t-1}|| \leqslant \epsilon/K_g, \epsilon \geqslant 0, \tag{4}$$

where $K_f$ and $K_g$ are Lipschitz constant of vision encoder $f_\theta$ and text encoder $g_\varphi$. Interestingly, LoRA freezes the old weights and introduces a small set of parameters that achieve the goal of learning new tasks with $||\theta - \theta^{t-1}|| \leqslant \epsilon/K_f$, $||\varphi - \varphi^{t-1}|| \leqslant \epsilon/K_g, \epsilon \geqslant 0$. In other words, the proposed LoRA integration is a simple way to learn the objective of Eq. (4).

**Empirical verification.** In Figure 3, we compare the performance of LoRA integration with fine-tuning, EWC (Kirkpatrick et al., 2017), ZSCL (Zhang et al., 2024) and Mod-X (Ni et al., 2023).

Figure 3: Fine-tuning results on Flickr30K compare prior CL methods based on (a) image-caption recall, (b) ImageNet zero-shot accuracy, (c) regularization loss, and (d) CLIP loss. Similar to existing regularization methods, LoRA reduces forgetting but sacrifices performance on new tasks. Regularization methods create a conflict between regularization loss and CLIP loss. In contrast, our method (C-CLIP) aligns the trends of these losses.

From the results, we can draw the following key observations: (1) While all the methods improve the performance on the current downstream dataset (Figure 3 (a)), EWC, ZSCL and Mod-X ruin the zero-shot performance on ImageNet-1K (Figure 3 (b)). (2) The proposed LoRA integration leads to less forgetting of zero-shot classification ability (Figure 3 (b)), but the performance on the fine-tuned dataset is undesirable (Figure 3 (a)). Therefore, in the following subsection 4.2, we propose a novel contrastive knowledge consolidation strategy to improve both plasticity and stability.

## 4.2 CONTRASTIVE KNOWLEDGE CONSOLIDATION FOR LEARNING ENHANCEMENT

In section 4.1, we have shown that LoRA integration can effectively reduce the forgetting of previous knowledge. However, similar to other existing methods, the performance on new datasets is undesirable. This is because simply aligning the new model with the old feature space naturally performs poorly for new datasets and ignores the characteristics of multimodal tasks. Is it possible to reduce forgetting and improve new task performance simultaneously? To achieve this goal, our high-level idea is *optimize CLIP to learn a better feature space from the old model, rather than just aligning with it*. Technically, each image-text pair $(\boldsymbol{v}_i^t, \boldsymbol{c}_i^t) \sim \mathcal{D}^t$ is mapped to deep feature space by both the old model $\{f_{\boldsymbol{\theta}^{t-1}}; \ g_{\boldsymbol{\varphi}^t}\}$ and new model $\{f_{\boldsymbol{\theta}^{t-1}}; \ g_{\boldsymbol{\varphi}^t}\}$, then we propose **contrastive knowledge consolidation** (CKC) which consists two important points:

- First, we introduce a projector $h_{\boldsymbol{\psi}} : \mathcal{Z} \to \mathcal{Z}$ after the vision and text encoders, optimizing the model in the projected space to keep the new and old feature spaces connected but not identical. This improves the plasticity for learning new tasks.

- Second, for each image-text pair, its feature instances in the old, projected space is viewed as positive while that of others are viewed as negative. This remarkably increases positive and negative pairs. Training CLIP to align with the old projected features and away from the irrelevant features improves new task performance and mitigates forgetting.

Figure 4 presents a comparison of CKC and regularization loss, and the CKC loss for the $t$-th incremental task is as follows:

$$\mathcal{L}_{\text{CKC}}^t = -\frac{1}{2N} \sum_{i=1}^{2N} \left( \log \frac{\exp\left(\widetilde{\boldsymbol{h}}_i^{t\,\top} \widetilde{\boldsymbol{z}}_i^{t-1}/\tau\right)}{\sum_{j=1}^{2N} \exp\left(\widetilde{\boldsymbol{h}}_i^{t\,\top} \widetilde{\boldsymbol{z}}_j^{t-1}/\tau\right)} + \log \frac{\exp\left(\widetilde{\boldsymbol{z}}_i^{t-1\,\top} \widetilde{\boldsymbol{h}}_i^t/\tau\right)}{\sum_{j=1}^{2N} \exp\left(\widetilde{\boldsymbol{z}}_i^{t-1\,\top} \widetilde{\boldsymbol{h}}_j^t/\tau\right)} \right), \quad (5)$$

$$\widetilde{\boldsymbol{h}}_i^t = \frac{[h_{\boldsymbol{\psi}}(f_{\boldsymbol{\theta}^t}(\boldsymbol{v}_i)), \ h_{\boldsymbol{\psi}}(g_{\boldsymbol{\varphi}^t}(\boldsymbol{c}_i))]}{\|[h_{\boldsymbol{\psi}}(f_{\boldsymbol{\theta}^t}(\boldsymbol{v}_i)), \ h_{\boldsymbol{\psi}}(g_{\boldsymbol{\varphi}^t}(\boldsymbol{c}_i))]\|}, \quad \widetilde{\boldsymbol{z}}_i^{t-1} = \frac{[f_{\boldsymbol{\theta}^{t-1}}(\boldsymbol{v}_i), \ g_{\boldsymbol{\varphi}^{t-1}}(\boldsymbol{c}_i)]}{\|[f_{\boldsymbol{\theta}^{t-1}}(\boldsymbol{v}_i), \ g_{\boldsymbol{\varphi}^{t-1}}(\boldsymbol{c}_i)]\|},$$

where $[,]$ denotes the concatenation operation, $N$ is the batch size, $\tau$ is the temperature parameter, and $\|\cdot\|$ denotes the Euclidean norm.

**Total training objective.** In addition, the original CLIP loss is used in the new feature space:

$$\mathcal{L}_{\text{CLIP}}^t = -\frac{1}{2N} \sum_{i=1}^{N} \left( \log \frac{\exp\left(\boldsymbol{z}_{v,i}^{t\,\top} \boldsymbol{z}_{c,i}^t/\tau\right)}{\sum_{j=1}^{N} \exp\left(\boldsymbol{z}_{v,i}^{t\,\top} \boldsymbol{z}_{c,j}^t/\tau\right)} + \log \frac{\exp\left(\boldsymbol{z}_{c,i}^{t\,\top} \boldsymbol{z}_{v,i}^t/\tau\right)}{\sum_{j=1}^{N} \exp\left(\boldsymbol{z}_{c,i}^{t\,\top} \boldsymbol{z}_{v,j}^t/\tau\right)} \right), \quad (6)$$

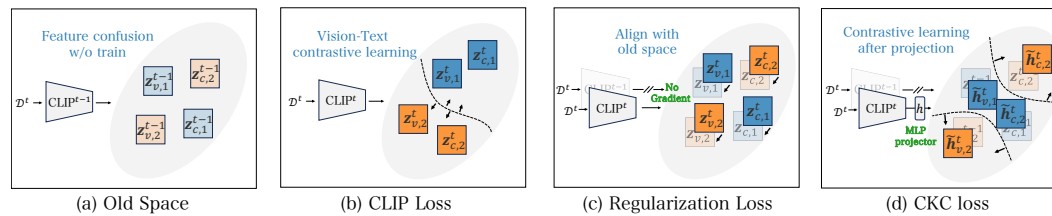

| (a) Old Space | (b) CLIP Loss | (c) Regularization Loss | (d) CKC loss |

Figure 4: Illustration of CLIP, CKC, and regularization losses: Previous methods align models to the old feature space, conflicting with CLIP optimization. CKC performs contrastive learning between old and new image-text samples in the projected space, aligning with CLIP loss.

where $z_{v,i}^t = \frac{f_{\theta^t}(v_i)}{\|f_{\theta^t}(v_i)\|}$ and $z_{c,i}^t = \frac{g_{\varphi^t}(c_i)}{\|g_{\varphi^t}(c_i)\|}$. The final loss is $\mathcal{L}^t = \mathcal{L}_{\text{CKC}}^t + \mathcal{L}_{\text{CLIP}}^t$. The total training process can be summarized as follows: At each continual stage $t$, the model is augmented with LoRA and optimized with $\mathcal{L}^t$; At the end of stage $t$, the LoRA module is integrated into the backbone based on Eq. 2 (we simply use $\alpha = 0.5$ for all experiments of C-CLIP). Figure 3(c) compares the loss curve of our method with other advanced CL methods. The results show that our loss curve closely follows CLIP's trend, significantly differing from previous methods.

## 5 EXPERIMENTS

Table 3: Comparison results of image-text retrieval on trained datasets. We continually fine-tune eight image-caption datasets, and then evaluate the performance after fine-tuning the final dataset.

| | Methods | flickr30k | COCO | pet | lexica | simpsons | wikiart | kream | sketch | average |
|---|---|---|---|---|---|---|---|---|---|---|
| **I2T R@1** | CLIP `ViT-B/16` | 35.80 | 10.40 | 1.96 | 3.99 | 2.62 | 4.40 | 9.41 | 1.70 | 8.79 |
| | EWC | 60.38 | 31.58 | 12.23 | 20.06 | 15.13 | 37.40 | 33.63 | 7.93 | 27.29 |
| | ZSCL | 66.79 | 37.15 | 13.34 | 24.99 | 20.98 | 40.67 | 39.23 | 6.88 | 31.25 |
| | Mod-X | 61.62 | 33.94 | 11.38 | 18.04 | 18.18 | 39.39 | 35.94 | 6.95 | 28.18 |
| | MOE-CL | 63.63 | 34.11 | 12.45 | 23.03 | 17.03 | 41.09 | 38.67 | 7.27 | 29.66 |
| | DKR | 63.60 | 28.98 | 11.04 | 21.74 | 19.05 | 38.02 | 37.92 | 7.06 | 28.43 |
| | C-CLIP | **84.40** | **56.92** | **19.73** | **42.65** | **25.43** | **45.89** | **42.07** | **9.55** | **40.83**(+9.58) |
| **T2I R@1** | CLIP `ViT-B/16` | 55.88 | 28.83 | 10.28 | 19.59 | 13.43 | 16.34 | 18.30 | 4.25 | 20.86 |
| | EWC | 60.30 | 31.16 | 13.91 | 20.14 | 15.52 | 28.34 | 30.34 | 12.23 | 26.49 |
| | ZSCL | 65.52 | 39.85 | 14.22 | 25.37 | 19.91 | 34.76 | 34.99 | 11.98 | 30.83 |
| | Mod-X | 61.16 | 35.37 | 13.01 | 21.84 | 17.24 | 30.98 | 28.62 | 10.66 | 27.36 |
| | MOE-CL | 63.17 | 40.31 | 15.45 | 24.09 | 16.36 | 35.78 | 34.04 | 11.02 | 30.03 |
| | DKR | 62.72 | 37.73 | 14.94 | 23.60 | 20.00 | 31.31 | 30.91 | 10.26 | 28.93 |
| | C-CLIP | **73.74** | **42.82** | **17.91** | **41.47** | **24.32** | **45.27** | **43.57** | **14.67** | **37.97**(+7.14) |

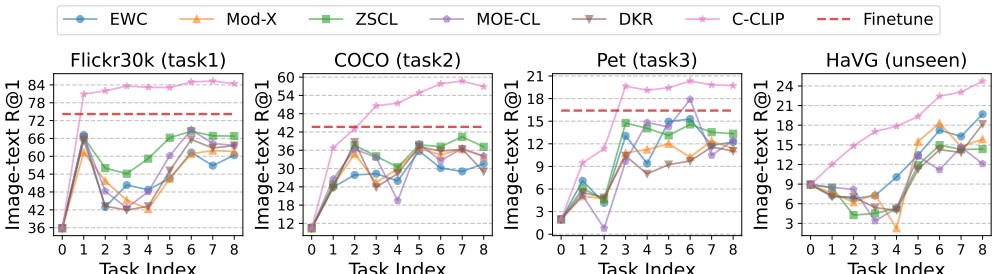

Figure 5: Continual fine-tuning performance on Flickr30k, COCO, Pets. HaVG is an unseen dataset.

**Networks and comparison methods.** We use CLIP (ViT-B/16) with pre-trained weights from large-scale open-world datasets as the backbone. It achieves a zero-shot classification accuracy of 67.73% on ImageNet-1K. The comparison methods include: (1) Classical CL methods like EWC (Kirkpatrick et al., 2017); (2) MTIL methods like ZSCL (Zhang et al., 2024) and MOE-CL (Yu et al., 2024), and (3) other recent methods like Mod-X (Ni et al., 2023) and DKR (Cui et al., 2024).

Table 4: Zero-shot accuracy of ImageNet-1K (ZS-I1K) and CIFAR-100 (ZS-C100) during continual fine-tuning on eight datasets. Our method maintained high zero-shot performance.

| | Method | Task ID | | | | | | | | | PD ($\downarrow$) |
| | | 0 | 1 | 2 | 3 | 4 | 5 | 6 | 7 | 8 | |
|---|---|---|---|---|---|---|---|---|---|---|---|
| **ImageNet** | EWC | 67.73 | 56.85 | 50.67 | 43.43 | 46.97 | 48.10 | 46.19 | 44.50 | 40.76 | 26.97 |
| | ZSCL | 67.73 | 59.10 | 56.51 | 52.47 | 51.74 | 51.03 | 50.89 | 50.02 | 49.60 | 18.13 |
| | Mod-X | 67.73 | 57.63 | 52.05 | 48.15 | 47.53 | 47.79 | 48.76 | 46.63 | 44.21 | 23.52 |
| | MOE-CL | 67.73 | 58.82 | 54.29 | 49.08 | 49.75 | 50.33 | 50.61 | 48.88 | 47.05 | 20.68 |
| | DKR | 67.73 | 58.54 | 53.69 | 49.55 | 47.80 | 47.55 | 49.67 | 45.46 | 45.88 | 21.85 |
| | C-CLIP | **67.73** | **63.11** | **65.31** | **63.26** | **63.31** | **61.95** | **62.13** | **61.51** | **60.31** | **7.42**$_{(+10.71)}$ |
| **CIFAR100** | EWC | 66.87 | 55.85 | 53.88 | 46.08 | 42.73 | 47.58 | 47.63 | 43.93 | 42.79 | 24.08 |
| | ZSCL | 66.87 | 58.70 | 55.06 | 51.28 | 47.37 | 50.92 | 52.65 | 53.02 | 52.19 | 14.68 |
| | Mod-X | 66.87 | 56.91 | 55.46 | 48.50 | 44.04 | 43.43 | 49.82 | 49.49 | 44.34 | 22.53 |
| | MOE-CL | 66.87 | 58.06 | 58.49 | 50.13 | 45.29 | 48.42 | 50.75 | 50.11 | 49.97 | 16.90 |
| | DKR | 66.87 | 57.06 | 56.72 | 49.00 | 44.49 | 42.15 | 48.09 | 48.44 | 46.45 | 20.42 |
| | C-CLIP | **66.87** | **63.17** | **64.78** | **64.08** | **61.02** | **60.55** | **62.85** | **62.02** | **61.58** | **5.29**$_{(+9.39)}$ |

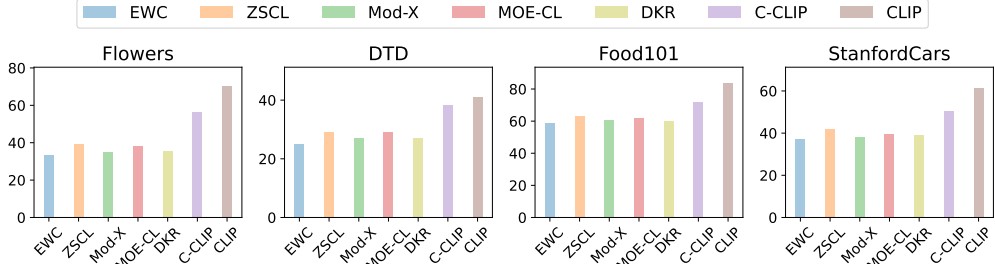

Figure 6: Zero-shot accuracy of Flowers, DTD, Food101, and Stanford Cars after continual fine-tuning on eight image-caption datasets. C-CLIP significantly outperforms previous CL methods.

**Implementation details.** C-CLIP is implemented in PyTorch lightning and trained on 8 NVIDIA 4090 GPUs with a batch size of 1024. Input images are resized to $224 \times 224$, and we train for 40 epochs on each dataset. The initial learning rate is set to $1 \times 10^{-6}$ with a 5-epoch warm-up using a cosine-decay learning rate scheduler. The low-rank decomposition ($R$) of LoRA is set to 16, with a scaling factor of $2 \times R$ and dropout of 0.1. We use the AdamW optimizer with $\beta_1 = 0.9$, $\beta_2 = 0.99$, and a weight decay of 0.2. Learning rates are adjusted per dataset; for example, on COCO-caption (Chen et al., 2015), the image encoder's learning rate is $5 \times 10^{-7}$, and the text encoder's is $4 \times 10^{-5}$. More details are provided in Appendix A.2.

## 5.1 MAIN RESULTS

**Comparison on trained datasets.** As shown in Table 3, the CLIP model pre-trained on large-scale datasets performs poorly on unseen datasets. This aligns with findings in (Zhang et al., 2024; Ni et al., 2023). However, full fine-tuning of these datasets may cause severe forgetting. For example, after fine-tuning twice on Flickr30K and COCO, the ImageNet zero-shot accuracy dropped from 67% to 25%. Previous CL methods show considerable gaps in new tasks when compared to full fine-tuning, sacrificing new task performance to mitigate forgetting. Our method significantly outperforms past approaches on new tasks. For instance, on datasets like Flickr30K and COCO, it greatly surpasses full fine-tuning in image2text retrieval performance. Moreover, As shown in Figure 5, C-CLIP improves performance on old tasks as new tasks are learned, which is a notable difference from previous CL methods. This demonstrates that our method can improve both new and old task performance during continual fine-tuning.

**Comparison on unseen datasets.** It is obvious that fine-tuning improves performance on the trained datasets. However, as shown in Figure 5, we observe that fine-tuning some task-specific datasets improves image-text retrieval performance on unseen datasets. Previous CL methods have shown a similar trend, but their training is unstable. For example, when fine-tuned on AI-generated datasets like Lexica, these methods suffer significant performance drops on real-world datasets like COCO and HaVG. This indicates that training on AI-generated datasets causes the model to forget its per-

Table 5: The effectiveness of each component in C-CLIP. The combination of LoRA and CKC results in better performance on new tasks with less forgetting.

| Method | flickr30k (task 0) | | | | COCO (task 1) | | | |
|---|---|---|---|---|---|---|---|---|
| | I2T R@1 | T2I R@1 | ZS-I1K | ZS-C100 | I2T R@1 | T2I R@1 | ZS-I1K | ZS-C100 |
| CLIP `ViT-B/16` | 35.80 | 55.88 | 67.73 | 66.87 | 10.40 | 28.32 | 67.73 | 66.87 |
| + Fine-tune | 74.20 | 76.00 | 47.46 | 49.67 | 43.26 | **44.86** | 23.87 | 25.65 |
| + LoRA | 66.90 | 73.10 | 62.18 | 62.53 | 38.17 | 40.55 | 61.49 | 61.03 |
| + CKC | **81.39** | **76.16** | 51.24 | 53.08 | **45.63** | 43.95 | 45.67 | 49.72 |
| + LoRA & CKC | 80.90 | 75.08 | **63.11** | **63.17** | 43.04 | 42.02 | **65.31** | **64.78** |

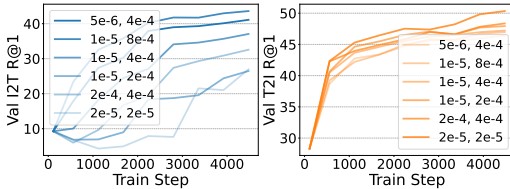

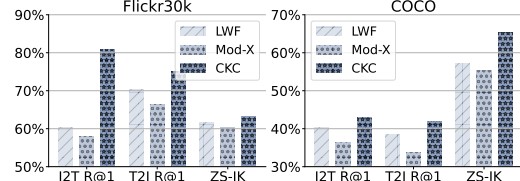

Figure 7: Impact of Learning rate: (left) image-to-text recall; (right) text-to-image recall.

Figure 8: Impact of Regularization losses on Flickr30K (left) and COCO-caption (right) .

formance in real-world domains. Our method exhibits impressive performance, when trained on AI-generated datasets in Task 4, the model also improves on unseen real-world datasets. This suggests that C-CLIP can continuously fine-tune using data from different domains.

**Zero-shot Classification ability.** Table 4 and Figure 6 verify that previous CL methods are useful for maintaining zero-shot ability, showing that their strategy of minimizing updates to feature space helps reduce forgetting (though it may impact performance on new tasks). Our method demonstrates the best anti-forgetting performance. For instance, after two tasks fine-tuning from Flickr30K to COCO, the ImageNet zero-shot accuracy remained at 65.1%, very close to the original 67.7%. When continuously fine-tuning 8 datasets, C-CLIP still maintained high zero-shot performance.

## 5.2 ABLATION STUDY AND FURTHER ANALYSIS

**Impact of components in C-CLIP.** As shown in Table 5, we evaluated the impact of each component in C-CLIP. LoRA significantly reduces forgetting but compromises new task performance, which is similar to previous CL methods. CKC improves both the learning ability for new tasks and reduces forgetting. However, CKC is not a strong constraint, so the forgetting remains relatively high. By combining CKC with LoRA, not only is forgetting further reduced, but new task performance matches or even exceeds that of full fine-tuning.

**Impact of learning rate.** Figure 7 shows results on COCO with varying learning rates. It can be seen that I2TR@1 is significantly affected by the learning rate. When the learning rates for the text encoder and image encoder are kept the same, CLIP training causes a degradation in the I2TR@1 metric. Therefore, we set the text encoder's learning rate to 80 times that of the image encoder to

Table 6: Comparison results of parameter count and model performance with different LoRA ranks ($R$).

| Method | Params | I2T R@1 | T2I R@1 | ZS-I1K | ZS-C100 |
|---|---|---|---|---|---|
| Fine-tune | 149M | 74.20 | 76.00 | 47.46 | 49.67 |
| $R = 8$ | **27.6M** | 80.42 | 74.81 | 63.05 | 63.13 |
| $R = 16$ | 29.1M | 80.90 | **75.08** | **63.11** | **63.17** |
| $R = 32$ | 32.0M | **80.95** | 75.01 | 63.03 | 63.15 |
| $R = 64$ | 37.9M | 81.11 | 75.03 | 62.88 | 63.09 |

achieve optimal results. Further configuration details can be found in Appendix A.2.

**Impact of regularization losses.** In Figure 8, we compare different regularization losses with CKC. Intuitively, these regularization methods, such as EWC, LWF, Mod-X, and LoRA, have a similar effect by reducing changes in feature space. However, experimental results show that these losses are not easily compatible with LoRA. When using LoRA, we reduced the coefficients of these regularization losses, but their performance remained poor, leading to worse new task performance and more severe forgetting. In contrast, CKC integrates much better with LoRA.

Table 7: The effectiveness of our method across various ViT backbones.

| Backbone | Method | flickr30k (task0) | | | | COCO (task1) | | | |
|---|---|---|---|---|---|---|---|---|---|
| | | I2T R@1 | T2I R@1 | ZS-I1K | ZS-C100 | I2T R@1 | T2I R@1 | ZS-I1K | ZS-C100 |
| ViT-B/32 | Vanilla | 31.10 | 52.94 | **62.59** | **61.24** | 8.30 | 25.53 | **62.59** | **61.24** |
| | C-CLIP | **79.34** | **73.05** | 58.72 | 58.93 | **42.57** | **41.46** | 59.95 | 59.61 |
| ViT-B/16 | Vanilla | 35.80 | 55.88 | **67.73** | **66.87** | 10.40 | 28.32 | **67.73** | **66.87** |
| | C-CLIP | **80.90** | **75.08** | 63.11 | 63.17 | **43.04** | **42.02** | 65.31 | 64.78 |
| ViT-L/14 | Vanilla | 59.10 | 61.66 | **75.01** | **76.78** | 25.86 | 33.14 | **75.01** | **76.78** |
| | C-CLIP | **92.59** | **79.40** | 70.56 | 70.90 | **64.22** | **51.63** | 72.18 | 72.24 |
| ViT-L/14@336px | Vanilla | 59.30 | 63.38 | **75.76** | **75.95** | 25.70 | 33.60 | **75.76** | **75.95** |
| | C-CLIP | **93.15** | **79.63** | 71.39 | 70.04 | **64.59** | **52.31** | 72.74 | 71.21 |

Table 8: Compare our method with other prompt-based continual learning approaches.

| | Method | Task ID | | | | | | | |
|---|---|---|---|---|---|---|---|---|---|
| | | 1 | 2 | 3 | 4 | 5 | 6 | 7 | 8 |
| ZS-I1K | LoRA | 62.18 | 61.49 | 58.47 | 57.74 | 55.03 | 53.89 | 52.02 | 51.02 |
| | L2P | 59.26 | 60.37 | 58.74 | 58.51 | 57.85 | 56.41 | 57.21 | 56.04 |
| | CPE-CLIP | 57.36 | 59.87 | 59.13 | 58.58 | 56.61 | 56.11 | 55.45 | 55.40 |
| | C-CLIP | **63.11** | **65.31** | **63.26** | **63.31** | **61.95** | **62.13** | **61.51** | **60.31** |
| I2T-T1 | LoRA | 66.90 | 60.83 | 55.71 | 58.04 | 61.85 | 59.85 | 62.29 | 63.28 |
| | L2P | 58.82 | 52.88 | 47.61 | 48.01 | 46.29 | 46.93 | 48.54 | 47.74 |
| | CPE-CLIP | 61.55 | 57.78 | 53.80 | 54.14 | 54.36 | 52.79 | 51.23 | 52.87 |
| | C-CLIP | **80.91** | **82.07** | **83.62** | **83.29** | **83.13** | **85.05** | **85.58** | **84.40** |
| I2T @R1 | LoRA | 63.28 | 38.91 | 11.32 | 28.69 | 18.94 | 37.89 | 34.37 | 6.63 |
| | L2P | 47.74 | 20.73 | 7.51 | 15.65 | 11.84 | 25.54 | 23.91 | 4.33 |
| | CPE-CLIP | 52.87 | 23.72 | 8.19 | 17.42 | 13.96 | 28.99 | 27.15 | 5.44 |
| | C-CLIP | **84.40** | **56.92** | **19.73** | **42.65** | **25.43** | **45.89** | **42.07** | **9.55** |

**Parameters and LoRA setting.** Table 6 compares the trainable parameters between full fine-tuning and various LoRA configurations. LoRA significantly reduces the number of trainable parameters, and most of the trainable parameters are embedding layers. As seen from the results of Task 0 and Task 1, different LoRA settings perform similarly in learning new tasks and preventing zero-shot forgetting. For simplicity, we set the LoRA rank to 16 in our experiments.

**Evaluation across ViT architectures.** In our main experiments, we use ViT-B/16 with a batch size of 1024. To demonstrate the effectiveness of C-CLIP across different architectures, we also test it on other backbones, including ViT-B/32, ViT-L/14, and ViT-L/14@336. Due to memory constraints, we reduce the batch size to 256 for the larger ViT-L models. As shown in Table 7, ViT-L significantly outperforms ViT-B, with substantial improvements in zero-shot performance on downstream datasets. Our method performs well across various ViT architectures, maintaining strong zero-shot capabilities and excelling in image-text retrieval on downstream tasks.

**LoRA vs. Prompt-tuning.** Prompt-based continual learning methods fix pre-trained models and only train prompts. We evaluate two representative methods, L2P (Wang et al., 2022) and CPE-CLIP (D'Alessandro et al., 2023). Table 8 reveals two key findings: (1) Prompt-tuning preserves general zero-shot performance on ImageNet-1K, whereas LoRA alone suffers from increased forgetting as tasks accumulate. (2) However, prompt-tuning does not prevent forgetting in downstream tasks, as updating prompts for new tasks leads to significant forgetting of prior ones—for instance, performance on the first task (I2T-T1) and all tasks (I2T@R1) is notably worse than LoRA after learning all eight tasks. In summary, while prompt-tuning preserves the original model's knowledge, it struggles to learn and retain new tasks. In contrast, combining LoRA and CKC proves to be more effective in this scenario.

## 6 CONCLUSION

This work focuses on the continual learning of visual-language models. We establish a multimodal continual learning benchmark and call for evaluating the performance from three different aspects. Then, we propose C-CLIP that prevents forgetting and enhances new task learning impressively with LoRA integration and contrastive knowledge consolidation. Comprehensive experiments demonstrate that the proposed C-CLIP outperforms existing state-of-the-art methods and achieves strong multimodal continual learning performance across image-text datasets from various domains.

## 7 ACKNOWLEDGMENTS

This work has been supported by the InnoHK program.

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

## A APPENDIX

### A.1 PROOF OF LIPSCHITZ CONTINUOUS

Let the function

$$f(w, x) = \sigma \left( w_n \cdot \sigma \left( w_{n-1} \cdot \sigma \left( \ldots \sigma \left( w_1 \cdot x \right) \right) \right) \right), \tag{7}$$

where $w = \{w_1, w_2, \ldots, w_n\}$, $x$ is an input vector, $w_i$ are weight matrices, and $\sigma$ is an activation function. Assume that the activation function $\sigma$ is bounded and Lipschitz continuous; that is, there exist constants $M_\sigma, L_\sigma > 0$ such that for all $z$,

$$|\sigma(z)| \le M_\sigma, \quad |\sigma(u) - \sigma(v)| \le L_\sigma |u - v|. \tag{8}$$

Also, the input $x$ and all weights $w_i$ have bounded norms; that is, there exist constants $M_x, M_w > 0$ such that

$$\|x\| \le M_x, \quad \|w_i\|_F \le M_w, \tag{9}$$

where $\| \cdot \|_F$ denotes the Frobenius norm. Under these conditions, we aim to prove that the function $f(w, x)$ is Lipschitz continuous with respect to the weights $w$; that is, there exists a constant $K > 0$ such that for any $w$ and $w'$,

$$\|f(w, x) - f(w', x)\| \le K \|w - w'\|_F. \tag{10}$$

To prove this proposition, First, consider the base case $n = 1$. In this case, the function simplifies to $f(w_1, x) = \sigma(w_1 x)$. For two weight matrices $w_1$ and $w_1'$, we have

$$\|f(w_1, x) - f(w_1', x)\| = \|\sigma(w_1 x) - \sigma(w_1' x)\|. \tag{11}$$

Since $\sigma$ is Lipschitz continuous,

$$\|\sigma(w_1 x) - \sigma(w_1' x)\| \le L_\sigma \|w_1 x - w_1' x\|. \tag{12}$$

Moreover,

$$\begin{aligned}
\|w_1 x - w_1' x\| &= \|(w_1 - w_1') x\| \\
&\le \|w_1 - w_1'\|_F \|x\| \\
&\le M_x \|w_1 - w_1'\|_F,
\end{aligned} \tag{13}$$

therefore,

$$\|f(w_1, x) - f(w_1', x)\| \le L_\sigma M_x \|w_1 - w_1'\|_F. \tag{14}$$

This shows that when $n = 1$, $f(w_1, x)$ is Lipschitz continuous with respect to $w_1$, with Lipschitz constant $K_1 = L_\sigma M_x$.

Next, assume that for $n = k$, the function $f_k(w_{1:k}, x)$ is Lipschitz continuous with respect to $w_{1:k} = \{w_1, w_2, \ldots, w_k\}$; that is,

$$\|f_k(w_{1:k}, x) - f_k(w_{1:k}', x)\| \le K_k \|w_{1:k} - w_{1:k}'\|_F. \tag{15}$$

Now, we need to prove that the conclusion holds for $n = k + 1$. For $n = k + 1$, the function is

$$f(w, x) = \sigma \left( w_{k+1} f_k(w_{1:k}, x) \right). \tag{16}$$

Considering two sets of weights $w = \{w_{1:k}, w_{k+1}\}$ and $w' = \{w_{1:k}', w_{k+1}'\}$, we have

$$\|f(w, x) - f(w', x)\| = \left\| \sigma \left( w_{k+1} f_k(w_{1:k}, x) \right) - \sigma \left( w_{k+1}' f_k(w_{1:k}', x) \right) \right\|. \tag{17}$$

Using the triangle inequality and the Lipschitz continuity of $\sigma$, we obtain

$$\begin{aligned}
\|f(w, x) - f(w', x)\| &\le L_\sigma \left( \left\| w_{k+1} f_k(w_{1:k}, x) - w_{k+1} f_k(w_{1:k}', x) \right\| \right. \\
&\qquad + \left. \left\| w_{k+1} f_k(w_{1:k}', x) - w_{k+1}' f_k(w_{1:k}', x) \right\| \right) \\
&= L_\sigma \left( \left\| w_{k+1} \left( f_k(w_{1:k}, x) - f_k(w_{1:k}', x) \right) \right\| \right. \\
&\qquad + \left. \left\| \left( w_{k+1} - w_{k+1}' \right) f_k(w_{1:k}', x) \right\| \right).
\end{aligned} \tag{18}$$

For the first term,

$$\begin{aligned}
\left\| w_{k+1} \left( f_k(w_{1:k}, x) - f_k(w_{1:k}', x) \right) \right\| &\le \|w_{k+1}\|_F \left\| f_k(w_{1:k}, x) - f_k(w_{1:k}', x) \right\| \\
&\le M_w K_k \|w_{1:k} - w_{1:k}'\|_F.
\end{aligned} \tag{19}$$

For the second term,

$$\left\| \left( w_{k+1} - w'_{k+1} \right) f_k(w'_{1:k}, x) \right\| \leq \| w_{k+1} - w'_{k+1} \|_F \| f_k(w'_{1:k}, x) \|$$
$$\leq \| w_{k+1} - w'_{k+1} \|_F M_f. \tag{20}$$

where $M_f$ is the boundedness constant of $f_k$. Combining the results above, we have

$$\| f(w, x) - f(w', x) \| \leq L_\sigma \left( M_w K_k \| w_{1:k} - w'_{1:k} \|_F + M_f \| w_{k+1} - w'_{k+1} \|_F \right). \tag{21}$$

Since

$$\| w_{1:k} - w'_{1:k} \|_F \leq \| w - w' \|_F, \quad \| w_{k+1} - w'_{k+1} \|_F \leq \| w - w' \|_F, \tag{22}$$

we can let

$$K_{k+1} = L_\sigma (M_w K_k + M_f), \tag{23}$$

thus obtaining

$$\| f(w, x) - f(w', x) \| \leq K_{k+1} \| w - w' \|_F. \tag{24}$$

This shows that when $n = k+1$, $f(w, x)$ is Lipschitz continuous with respect to $w$. By mathematical induction, we conclude that for any $n$, the function $f(w, x)$ is Lipschitz continuous with respect to the weights $w$. ∎

(a) Pets

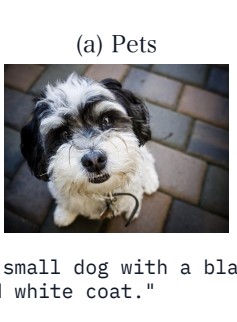

"a small dog with a black and white coat."

(b) Kream

"outer, Patagonia Classic Retro-X Jacket Natural, a photography of a tan jacket with a blue pocket."

(c) Simpsons

"A man dressed as Santa Claus stand next to a woman."

(d) Wikiart

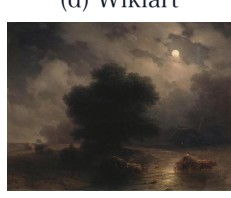

"A dark landscape with a bright moon and a large central tree."

(e) Sketch

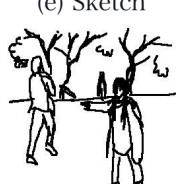

"People are playing and walking around in a park."

(f) Lexica

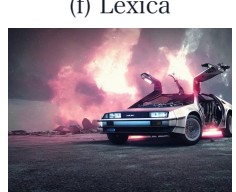

"ultra realistic delorean dmc 5 drifts on road wreckag..."

Figure 9: Examples of image-text data from different domains in the VLCL benchmark.

## A.2 ADDITIONAL EXPERIMENTAL DETAILS

In Figure A.2, we present image-text data from different domains in the VLCL benchmark. Some image caption datasets have predefined splits, such as Flickr30K and COCO-caption, with test sets of 1K and 5K, respectively. For Pet, Lexica, and HausaVG, we evaluate their test sets. For other datasets like Simpsons, Sketch, and Wikiart, we randomly split 80% for training and 20% for testing. For Kream, the training and test sets are evenly divided. Figure 3 shows examples of tasks from different domains in the VLCL benchmark. During training, all images are resized to 224x224, and the maximum text length is set to 77.

Table 9: Comparison of training time (Epoch/s).

| Method | flickr30k | COCO |
|---|---|---|
| Fine-tune | 13.15 | 44.58 |
| + LWF | **15.72** | **53.20** |
| + Mod-X | 16.91 | 57.13 |
| + CKC | 16.59 | 56.07 |

Regarding learning rates, we tested several values. For COCO-caption, we set the learning rate for the text encoder to 80 times that of the image encoder, while for other datasets, it was set to 10 times.

The base learning rate was 5e-7 for COCO-caption, 1e-5 for Flickr30K, and 3e-5 for other datasets, from Pet to Sketch. Our method's code will be open-sourced, and further details can be found in the released code.

### A.3  TRAINING EFFICIENCY COMPARISON

We assess the efficiency of CKC and other regularization losses by comparing their average per-epoch training times over 40 epochs. As shown in Table 9, the additional losses introduce some overhead. For instance, on COCO, LWF and Mod-X increased training time by 19.3% and 28.2%, respectively. The computational overhead of our method is comparable to these regularization losses, leading to a 25.2% increase in training time.

### A.4  LIMITATION

Our method is mainly designed for contrastive learning based version-language models like CLIP. More advanced multimodal large language models like LLaVA (Liu et al., 2024) typically use language modeling autoregressive loss. Nevertheless, considering the version encoder in LLaVA is from CLIP and plays an important role in LLaVA, we believe the proposed C-CLIP can also enhance the vision ability of multimodal large language models. In the future, we will explore continual learning for generative VLMs like LLaVA.

