# OpenReview forum: "C-CLIP: Multimodal Continual Learning for Vision-Language Model"
_ICLR.cc/2025/Conference — ICLR 2025 Poster_

### Official Review · Reviewer_45wV · 2024-10-31

**Soundness:** 3
**Presentation:** 4
**Contribution:** 3
**Rating:** 6
**Confidence:** 5

**Summary:**

The paper introduces a rich benchmark for general vision-language continual learning focused on evaluating performance degradation in zero-shot generalization as well as class-incremental performance. The paper also introduces a modified CLIP model to account for some limitations of similar models in the literature. The proposed model, C-CLIP, uses LoRA adaptation and Contrastive Knowledge Consolidation as the two main advancements of the standard CLIP backbone. The proposed model outperforms current SOTA models in preserving knowledge and zero-shot performance degradation.

**Strengths:**

Overall, I positively evaluate the proposed manuscript. I like the idea of proposing a complete evaluation benchmark encompassing several strategies for measuring zero-shot generalization capabilities in continual learning settings. The benchmark is rich enough to increase the challenge that continual learning literature normally faces when proposing novel models evaluated on limited and strict benchmark datasets.

**Weaknesses:**

(1) The introduction needs a significant extension. Authors keep claiming that vision-language continual learning is under-explored, but they missed relevant research papers. Some examples are:

- Qian et al (2023). Decouple before interact: Multi-modal prompt learning for continual visual question-answering
- Wang et al (2021). Continual learning in cross-modal retrieval
- D'Alessandro et al (2023). Multimodal Parameter-Efficient Few-Shot Class Incremental Learning
- Srinivasan et al (2022). Climb: A continual learning benchmark for vision-and-language tasks

just to name a few. I would also mention the general literature in FSCIL (Few-Shot Class-Incremental Learning) as well. All these works challenge vision-language continual learning from different perspectives and provide insightful feedback on specific problems encountered in multimodal settings.
Furthermore, contrastive learning pre-training offers more nuances compared to a standard unimodal continual learning scenario, since it can solve image classification as well as pure multimodal tasks, as mentioned in the main paper. But it's clear that the burden of solving forgetting and adaptability problems is totally different in the two tasks, since in the former the language encoder needs to perform representational learning only on fixed text labels to pair with images. This consideration comes directly when considering the papers already mentioned in the current introduction and the ones I mentioned before. The proposed one is a comprehensive benchmark, but the introduction needs to clarify these concepts.

(2) I think the overall benchmark definition is confused and badly explained. First of all, the authors want to clarify the role of each benchmark evaluation aspect. Some datasets are used in a continual learning fashion, with incremental experiences, others are used without the temporal factor, and you have to reach section 5.2 to figure it out. This is a benchmark that, I assume, the authors want to propose to unify evaluation criteria across VL CL models. I suggest putting a section in the main text (a table if it is necessary) clarifying how the benchmark is made, by referring to the continual learning datasets stats (e.g. number of splits, classes per split) and datasets role (e.g. continual learning, zero-shot generalization not in continual learning scenario). If there are space issues, I think there are sections that can be sacrificed since clarifying the benchmark is crucial (e.g. "Theoretical Analysis" section can be safely moved into the appendix).

(3) Since the proposed model uses a parameter-efficient strategy, I think it's worth mentioning the whole corpus of rehearsal-free continual learning methods based on parameter-efficient tuning (e.g. prompt tuning, prefix tuning) other than standard methods like regularization-, architecture-, and replay-based ones.

(4) Is the CLIP (first line) in Table 2 fine-tuned or just frozen? In Figure 3 there is a reference to "Fine-Tune" so I guess it's fine-tuned CLIP. If this is the case, why not add frozen CLIP in all the model comparisons as a baseline? It can be helpful to verify if pre-training alone is enough to ensure an acceptable generalization and performance degradation.

(5) Line 232. "the model has no access to the task identity". It's ok but please state from the beginning that we are in a class-incremental learning scenario and thus that the benchmark is for CIL, as usual in any CL paper.

(6) Line 255. "leran" -> "learn".

(7) "single-modal scenario" is an unpopular term. I think the literature agrees with using the term "unimodal" instead.

**Questions:**

I think that LoRA can be compared with other parameter-efficient methods that keep the main backbone frozen like prompt-tuning (e.g. Khattak et al (2022). MaPLe: Multi-modal Prompt Learning), unless there is a particular reason with a mathematical justification for keeping LoRA. But in my opinion, eq. 4 holds with any other non-adapters parameter-efficient methods, and CKC is not constrained by the backbone adaptation. I understand that the authors might complain that LoRA is just a design choice, as any other else, but the paper lacks a clear justification for that since eq. 2 and eq. 4 hold for prompt-tuning too, and the intention here seems just preserving knowledge by freezing the backbone. CKC seems a clearer design choice on the other hand.

Do you agree with this view, or am I missing something and maybe LoRA is already justified so there is no need to explore other parameter-efficient methods?

---

> ### Author Response · Authors · 2024-11-24
> **Reply to Reviewer 45wV (Part 1/2)**
>
> We sincerely thank Reviewer 45wV for your review and are grateful for the time you spent on our submission. We are glad for the acknowledgment that the proposed benchmark is rich and our method is novel. Below we would like to give detailed responses to each of your comments.
>
> **Q1. The introduction needs a significant extension.**
>
> Thank you for bringing those valuable studies to our attention. We have added them to the introduction section of the revised version, for a comprehensive review of relevant research papers.
> In the revised version of our paper, we have reorganized the existing challenges into three main sections:
>
> - Compared to standard unimodal continual learning scenarios, multimodal settings present greater complexity.
> - The evaluation of vision-language models (VLMs) remains insufficient.
> - Traditional CL methods apply regularization to reduce forgetting, which hinders the learning of new tasks.
>
> We hope the extended introduction aligns with your suggestions. It reorganizes challenges and provides a clearer structure to emphasize the unique contributions of our work.
>
>
>
> **Q2. I think the overall benchmark definition is confused and badly explained.**
>
> Thanks for this comment. Following your suggestion, we have thoroughly reorganized the description of the benchmark, added a new Table 2, and refined the content in section 5.2. Here the is updated description of the benchmark (excluding the citations of datasets for the Markdown presentation):
>
> VLCL benchmark. To evaluate the continual learning performance of vision-language models, we establish a novel benchmark that includes three evaluation tracks, summarized in Table 2.
> (1) Multimodal continual learning. Eight image-caption datasets are used in this track. Among them, Flickr30K and COCO-Caption are general real-world datasets. Other datasets, including Pets, Lexica, Simpsons, WikiArt, Kream, and Sketch, represent specific domains such as AI-generated images, art, clothing, illustrations, and sketches.
> (2) Zero-shot retrieval. One held image-caption dataset, HAVG, is used to assess retrieval performance on unseen datasets.
> (3) Zero-shot classification. Previous CL work on CLIP has overlooked an important aspect: the forgetting of general representations (i.e., zero-shot generalization). To evaluate this, we tested zero-shot performance on six image classification datasets: ImageNet, CIFAR-100, StanfordCars, Flowers, DTD, and Food101.
>
> In the revised paper, we clarified an important point: previous CL work often divides a single dataset into multiple incremental tasks, requiring a detailed explanation of the dataset partitioning. However, such a setting does not exist in our method, as we treat each dataset as a separate task.
>
>
> **Q3. It's worth mentioning the whole corpus of rehearsal-free continual learning methods based on parameter-efficient tuning.**
>
> Following your suggestions, we added rehearsal-free continual learning methods based on parameter-efficient tuning in section 2.1 on Page 2. Besides, to further demonstrate the strength of our method, we conduct experiments to compare our method with prompt-based CL methods such as L2P and CPE-CLIP, and added a new Table 8 in the revised paper. For L2P, we use a pool of 40 prompts (length 5) and append 5 prompts at the input. For CPE-CLIP, we add 2 prompts per layer, following the original configuration.
> Results in Table 8 reveal the following observations:
> - Prompt-tuning maintains general zero-shot performance in ImageNet-1K accuracy. While only using LoRA leads to forgetting general zero-shot knowledge as tasks increase.
> - Prompt-tuning does not eliminate forgetting on downstream tasks. Updating prompts for new tasks causes obvious forgetting of prior tasks, e.g., the performance on the first task (I2T-T1) and all tasks (I2T@R1) is much worse than LoRA after learning all eight tasks.
>
> In summary, although prompt-tuning exhibits less forgetting on the original model, it learns little from new tasks and forgets newly acquired knowledge easily when the model underperforms on downstream tasks. The combination of LoRA and CKC proves to be more effective in this scenario.

---

> ### Author Response · Authors · 2024-11-24
> **Reply to Reviewer 45wV (Part 2/2)**
>
> **Q4. Is the CLIP (first line) in Table 2 fine-tuned or just frozen?**
>
> The CLIP (first line) in Table 2 is just frozen, and the results are zero-shot performance. We have made this clear in the revised version.
>
>
> **Q5. Please state from the beginning that we are in a class-incremental learning scenario and thus that the benchmark is for CIL, as usual in any CL paper.**
>
> Thanks for this comment. Compared to the well-defined category-based continual learning in image classification, we focus on multimodal continual learning for VLM like CLIP, and view each dataset as one task, similar to the domain-incremental learning in classical CL or capability continual learning. The task ID information is not needed at inference time. We have made this clear on Page 4 (Evaluation metric part) in the revised version.
>
>
> **Q6. Line 255. "leran" -> "learn".**
>
> Thanks. We have corrected those typos and tried our best to check and correct other typos.
>
>
> **Q7. "single-modal scenario" is an unpopular term. I think the literature agrees with using the term "unimodal" instead.**
>
> Good suggestion. We have modified this point in the revised version following your suggestion.

---

### Official Review · Reviewer_L3rn · 2024-10-31

**Soundness:** 3
**Presentation:** 2
**Contribution:** 3
**Rating:** 6
**Confidence:** 4

**Summary:**

- This paper addresses continual learning scenarios for vision-language models, focusing on cross-modal retrieval across seen and unseen datasets.
- The proposed approach combines two key components: (1) low-rank adaptation (LoRA) training with theoretical analysis of its connection to continual learning objectives, and (2) contrastive knowledge consolidation (CKC), which distills knowledge from previous task models as an auxiliary loss.
- Experimental analysis is conducted with nine vision-language datasets, including one unseen dataset, where the proposed C-CLIP consistently outperforms competitive baselines. Additional experiments on zero-shot image classification and ablation studies further validate the method's effectiveness.

**Strengths:**

- The application of continual learning to vision-language scenarios addresses important practical needs.
- The proposed method effectively combines LoRA with knowledge distillation for vision-language continual learning, which sounds reasonable and is supported by theoretical analysis.
- The paper presents strong experimental results across a comprehensive range of benchmarks, demonstrating consistent performance improvements over baselines and thorough evaluation across different experimental setups.

**Weaknesses:**

- Delivery should be significantly improved. Here is a non-exhaustive list I have identified while reading the paper, but I believe there are plenty of points that need to be improved:
	- What CIL and MTIL stands for? (I don't find Table 1 to be really necessary, though.)
	- Figure 2 does not effectively convey what CKC is about.
	- The range covered by the y-axis in Figure 3-(c) seems too wide, making it hard to see what is happening locally. Consider normalizing and clamping. Also, it would be interesting to see if such a behavior, i.e., alignment in main and auxiliary losses, is universally observed in other tasks.
	- What does Figure 3 in L284 refer to?
	- Please assign proper variables for equation 5 and 6, instead of using $z$ everywhere. This makes the equations somewhat confounding, e.g., substituting $t$ in  $\tilde{z}_i^t$ with $t-1$ is not identical to $\tilde{z}_i^{t-1}$, etc.
	- Many typos, e.g., space is view as (L280), loss is use in the (L294), etc.
- While performance improvements in all major experiments are impressive (compared to performance gap among prior arts), it is hard to tell whether these results are reproducible and rigorously analyzed.
	- The codes are not provided, and the details in the paper may not be sufficient to fully reproduce the experiments reported.
	- Some important hyperparameters are not tested properly, e.g., $\alpha$ for LoRA integration, ratio of CKC loss and CLIP loss, etc. The performance gaps in Table 4 for each row is quite large, making it hard to understand at which point these improvements emerge.
	- Only a subset of tasks are reported in major figures, e.g., Figure 3, 4, 6, 7 and Table 4, 5. It would be beneficial to see if a similar tendency holds for the rest of the tasks during training end-to-end (at least for some crucial setups like ablation studies).
- Distillation in continual learning is not new, which was addressed in Contrastive Continual Learning (ICCV 2021) to the best of my knowledge. It would be necessary to provide some theoretical or empirical comparison between CKC and previous lines of work leveraging similar concepts.

**Questions:**

Please refer to the weaknesses section for major questions. Here are some minor comments about the paper:
- Eq. 23 is recursive with respect to K, could this bound serve as a meaningful bound? (could be considerably large for some nontrivial L and M?)
- L96 - cite correct paper (pioneering works, not recent ones)
- Is $h_\psi$ in L277 used elsewhere after training? Why is an identical projector used for both vision and text embeddings in equation 5?

---

> ### Author Response · Authors · 2024-11-24
> **Reply to Reviewer L3rn (Part 1/2)**
>
> We sincerely thank Reviewer L3rn for your review and are grateful for the time you spent on our submission. We are glad for the acknowledgment that the problem is practically important, the proposed method is effective and reasonable, and the experiment is strong. We appreciate the detailed suggestions for improving the presentation of the paper. Below we would like to give detailed responses to each of your comments.
>
>
> **Q1. Delivery should be significantly improved.**
>
> Thank you for your careful reading. We revised the submission following your comments and tried our best to check and correct other confusing descriptions and typos.
>
> **(1) What CIL and MTIL stand for? (I don't find Table 1 to be really necessary, though.)**
>
>  CIL denotes class-incremental learning, and MTIL denotes multi-domain task incremental learning (MTIL) benchmark, which is proposed by Zheng et al., (2023) and followed by Yu et al. (2024). Following your suggestion, we provide a clear explanation of CIL and MTIL in section 2.1. We have added a new table, Table 2, below Table 1 to provide a clearer comparison between the VLCL benchmark and previous CL benchmarks. These tables aim to highlight the distinctions of the VLCL benchmark.
>
> **(2) Figure 2 does not effectively convey what CKC is about.**
>
> Thanks for your comment. We provide a new Figure 4 to illustrate the proposed CKC in the revised version. Compared with other methods, CKC differs primarily in two aspects:
>
> - CKC introduces a projector $h_{{\psi}}: \mathcal{Z} \rightarrow \mathcal{Z}$ after the text encoder and optimize the model in the projected space.
>
> -  CKC optimizes CLIP to learn a better feature space from the old model, rather than aligning with it. This is achieved by increasing positive and negative pairs for contrastive learning: the relevant old feature instances of each image-text pair are treated as positives, while those of other pairs are treated as negatives. This optimization learns new knowledge and prevents forgetting simultaneously.
>
> We hope the newly added Fig. 4 illustrates the differences between CKC and previous regularization methods, and explain why the CKC loss can be optimized alongside the CLIP loss.
>
>
> **(3) The range covered by the y-axis in Figure 3c seems too wide,..,if is universally observed in other tasks.**
>
> Thanks for this suggestion. We have improved Fig. 3c to illustrate more details, showing that the CKC loss and CLIP loss exhibit the same downward trend during training.  Of course, this phenomenon is consistent across all tasks.
>
> **(4) What does Figure 3 in L284 refer to?**
>
> In our previous submission, Figure 3 in L284 refers to Figure 3c, which compares the training loss curves among our method and other regularization losses. In the revised version, we modify it to the newly provided figure 4.
>
>
> **(5) Please assign proper variables for equations 5 and 6, instead of using z  everywhere.**
>
> Good suggestion. We have modified the $\widetilde z^{t}_i$ as $\widetilde h^{t}_i$, which denotes the prejected features.
>
> **(6) Many typos, e.g., space is view as (L280), loss is use in the (L294), etc.**
>
> Thank you again for your careful review. We have corrected those typos and thoroughly checked the revised paper for any other errors.
>
>
> **Q2. While performance improvements in all major experiments are impressive, it is hard to tell whether these results are reproducible and rigorously analyzed.**
>
> We promise to release the source code and model checkpoints. Additionally, our method is simple and modular, the LoRA integration and CKC loss are easy to implement. In the following, we respond to each of your concerns in detail.
>
> **(1) The codes are not provided, and the details in the paper may not be sufficient to fully reproduce the experiments reported.**
>
> The code of this work (the proposed benchmark and our method) is promised to be publicly available once the work is published. Implementation details are provided in Section 5, and we are glad to share any additional details about our method.

---

> ### Author Response · Authors · 2024-11-24
> **Reply to Reviewer L3rn (Part 2/2)**
>
> **(2) Some important hyperparameters are not tested properly, e.g.,
>  for LoRA integration, the ratio of CKC loss and CLIP loss, etc.**
>
> In our work, for all experiments with C-CLIP, we simply set $\alpha=0.5$ , and it performs well. Our method introduces no additional hyperparameters for balancing the CKC loss and CLIP loss. Specifically, the CKC loss is formulated in a contrastive manner, naturally aligning with the CLIP loss. We have re-emphasized these points in the revised version. We aim for C-CLIP to be a simple yet solid method, with the important advantage of introducing no new hyperparameters to the vanilla CLIP model.
>
>
> **(3) Only a subset of tasks are reported in major figures, e.g., Figure 3, 4, 6, 7 and Table 4, 5. It would be beneficial to see if a similar tendency holds for the rest of the tasks during training end-to-end (at least for some crucial setups like ablation studies).**
>
> Therefore are eight tasks during continual learning, and we conducted ablation studies in two datasets that belong to the first two tasks to demonstrate the effectiveness of each component of our method.
> A key characteristic of continual learning is that the model learned at stage $t$ serves as the initialization for stage $t+1$. In previous CL methods, the performance differences between methods are often small on the initial tasks but grow significantly as more incremental tasks are introduced.
> Therefore, we present ablation experiments on the first two tasks. During subsequent learning sessions, the initial performance is already strong, making the impact of each component more pronounced as learning progresses.
>
>
>
> **Q3. Distillation in continual learning was addressed in Contrastive Continual Learning (ICCV 2021). It would be necessary to provide a comparison between CKC and previous lines of work leveraging similar concepts.**
>
> Thanks for bringing this work to our attention. We have added the citation in the revised version. The Co2L method regulates the sample similarity matrix, a regularization strategy also adopted in recent continual VLM works, such as Mod-X (ICML 2023) and DKR (AAAI 2024). Experiments show that our method outperforms this strategy.
> We believe that constraining the sample similarity matrix is not ideal because new datasets are often not well-distributed in the old feature space, leading to a trade-off between learning new knowledge and preserving the old feature space.
> In contrast, the proposed CKC loss, which has a formulation similar to the CLIP loss and is well-aligned with it (as demonstrated in Figure 3), enables simultaneous learning of new and old knowledge.
>
>
>
> **Q4. Eq. 23 is recursive with respect to K, could this bound serve as a meaningful bound? (could be considerably large for some nontrivial L and M?)**
>
> Thank you for pointing this out. Because Eq. (23) is recursive, the Lipschitz constant $K$ can grow exponentially with the number of layers if $L_\sigma M_w > 1$, making the bound potentially very large and less meaningful. To ensure the bound remains practical, we need to keep $L_\sigma M_w \leq 1$, for example, selecting activation functions with $L_\sigma \leq 1$ and controlling the weight norms $M_w \leq 1$.
>
> **Q5. L96 - cite the correct paper (pioneering works, not recent ones)**
>
> Thank you for this suggestion. These two recent ones are survey papers on continual learning. Following your suggestion, we cite pioneering works like EWC in the revised version.
>
> **Q6.  $h_{\psi}$ in L277 used elsewhere after training? Why is an identical projector used for both vision and text embeddings in equation 5?**
>
> Thank you for your comments. Below is our explanation regarding the CKC projector:
>
> - The projection layer $h_{\psi}$ in Line 277 is only used to compute the CKC loss. After training, C-CLIP and CLIP retain the same structure, with LoRA modules integrated into the model via reparameterization, and the projection layer $h_{\psi}$ is discarded.
> - The vision encoder and text encoder share a single projector for the CKC loss because the text and vision embeddings in CLIP are already aligned by the two built-in projection layers in CLIP itself. These existing projection layers in CLIP align the outputs of the vision and text encoders. CKC projector works in the text-vision aligned space, a single shared projector is sufficient.

---

> > ### Comment · Reviewer_L3rn · 2024-11-26
> >
> > Thank you for your thorough responses. The delivery seems much better now. I acknowledge that $\alpha$ and the balance between loss terms are reasonable enough as of now, but these are indeed hidden parameters that could affect the end performance, which are not tested for simplicity. How sensitive is the performance of each task to changes in $\alpha$? Which of the two loss terms is more important and shows higher sensitivity in the performance? I think these questions are important for rigorously understanding how the proposed framework operates.
> >
> > That being said, here are some minor comments:
> > - Shouldn't the CKC in Figure 2 be unidirectional (i.e., from m-1 to m)?
> > - It would be better to explicitly specify that $D_{m-1}$ in Figure 2 and its connected modules are from previous task(s).
> > - Since my question was initially about whether each graph in Figure 3-(c) is gradually diverging or converging, I don't think it is necessary to highlight early-stage dynamics, which are somewhat evident without highlighting.
> > - Do $L$ and $M$ of the model satisfy the practical constraints specified in the response?
> > - If experiment logs for the last six tasks for Figure 3, 4, 6, 7 and Table 4, 5 are stored internally, they could preferably be included in the Appendix as a reference.

---

### Official Review · Reviewer_8g5w · 2024-11-01

**Soundness:** 3
**Presentation:** 2
**Contribution:** 3
**Rating:** 8
**Confidence:** 3

**Summary:**

This paper presents C-CLIP, a new continual learning framework that aims to reduce the forgetting in Vision-Language Models (VLMs) while achieving high fine tuning accuracy on new datasets, specifically focused on CLIP. The authors also introduce a benchmark for Vision-Languge Continual Learning to evaluate CLIP’s performance in retaining zero-shot capabilities while acquiring new domain knowledge. To achieve that goal, C-CLIP combines Low-Rank Adaptation (LoRA) to reduce forgetting and Contrastive Knowledge Consolidation (CKC) to retain old knowledge. Experiments show that C-CLIP achieves high accuracy in downstream tasks while maintaining zero-shot classification performance across different domains.

**Strengths:**

The paper shows an innovative approach to continual learning within Vision-Language models, by focusing on the adaptation of new domains while retaining the zero-shot capabilities, with some key strengths:
- The integration of LoRA adapters and CKC is a good approach to achieving robust continual learning in VLMs, especifically capable of retaining zero-shot capabilities
- Experiments are performed with evaluations across a wide range of both seen and unseen domains, which highlight the method's effectiveness in preventing catastrophic forgetting.
- C-CLIP advantages against other CL methods are clearly demonstrated through detailed figures and tables, facilitating the understanding of the experimental results.
- A method that uses a Vision-Language model like CLIP in multimodal settings is highly relevant given the actual trend of focusing just on unimodal CL scenarios.

**Weaknesses:**

The paper also shows some weaknesses that could improve the overall result:
- While LoRA reduces significantly trainable parameters, it would be interesting and beneficial to include an analysis of the computational costs involved in using CKC during the fine tuning process.
- Although comparisons are made against baseline and some recent CL approaches, there is no comparison against other well known parameter efficient CL methods (e.g., DualPrompt, CPE-CLIP, Learning to Prompt, etc) that could strengthen the evaluation.

**Questions:**

- Could the C-CLIP approach be generalized to VLMs beyond CLIP, or are specific adjustments required?
- Could the authors further elaborate on why CKC avoids the trade-offs associated with other regularization methods?

---

> ### Author Response · Authors · 2024-11-24
> **Reply to Reviewer 8g5w**
>
> We sincerely appreciate Reviewer 8g5w for the positive recommendation as well as the valuable suggestions. We really appreciate your kind words that our work is innovative with key strengths. Below, we would like to give detailed responses to each of your comments.
>
>
> **Q1. Analysis of the computational costs involved in using CKC during the fine-tuning process.**
>
> Thanks for this suggestion. We have added the results in Appendix A.3 of the revised version. As shown in Table 9, the additional losses introduce some overhead. For instance, on COCO, LWF and Mod-X increased training time by 19.3% and 28.2%, respectively. The computational overhead of CKC is similar to these regularization losses, leading to a 25.2% increase in training time.
>
> **Q2. Comparison against other well-known parameter-efficient CL methods that could strengthen the evaluation.**
>
> Good suggestion. In Table 8 of the revised paper, we compare our method with prompt-based CL methods such as L2P and CPE-CLIP. For L2P, we use a pool of 40 prompts (length 5) and append 5 prompts at the input. For CPE-CLIP, we add 2 prompts per layer, following the original configuration.
> Results in Table 8 reveal the following observations:
> - Prompt-tuning maintains general zero-shot performance in ImageNet-1K accuracy. While only using LoRA leads to forgetting general zero-shot knowledge as tasks increase.
> - Prompt-tuning does not eliminate forgetting on downstream tasks. Updating prompts for new tasks causes obvious forgetting of prior tasks, e.g., the performance on the first task (I2T-T1) and all tasks (I2T@R1) is much worse than LoRA after learning all eight tasks.
>
> In summary, although prompt-tuning exhibits less forgetting on the original model, it learns little from new tasks and forgets newly acquired knowledge easily when the model underperforms on downstream tasks. The combination of LoRA and CKC proves to be more effective in this scenario.
>
>
>
> **Q3. Could the C-CLIP approach be generalized to VLMs beyond CLIP, or are specific adjustments required?**
>
> C-CLIP mainly consists of LoRA integration and constrastive knowledge consolidation (CKC). **(1)** The LoRA integration technique can be applied to any VLM such as BLIP, BLIP2, or InstructBLIP, as well as more advanced multimodal large language models (MLLM) such as LLaVA and miniGPT4. **(2)** The CKC can be directly applied to any contrastive training-based model. For multimodal language models that use language modeling loss, it is difficult to adapt CKC to them.
> However, since almost all VLM and MLLM leverage CLIP encoder, therefore the proposed C-CLIP can be used to enhance the version-language alignment ability of MLLMs. Inspired by your valuable suggestion, we added a discussion of this point in the revised paper, and will consider the continual learning for LLaVA in future work.
>
>
> **Q4. Could the authors further elaborate on why CKC avoids the trade-offs associated with other regularization methods?**
>
> Thank you for this comment. Specifically, other regularization methods typically align the new model with the old feature space. However, for new tasks, the old feature space could perform poorly,  forcing the model to trade off between learning new knowledge and preventing forgetting. In our method:
>
> -  We introduce a projector $h_{{\psi}}: \mathcal{Z} \rightarrow \mathcal{Z}$ after the vision and text encoders, optimizing the model in the projected space. In this way, the new and old feature spaces keep connected but are not identical, which improves the plasticity for learning new tasks while also maintaining the previous knowledge.
>
> -  CKC optimizes CLIP to learn a better feature space from the old model, rather than aligning with it.
> This is achieved by increasing positive and negative pairs for contrastive learning: the relevant old feature instances of each image-text pair are treated as positives, while those of other pairs are treated as negatives.  This optimization learns new knowledge and prevents forgetting simultaneously, thereby avoiding trade-offs.
>
> We have added a new Figure 4 in the revised paper to illustrate CKC. We hope that Figure 4 helps to clearly demonstrate the differences between CKC and previous regularization methods, as well as explain why the CKC loss can be effectively optimized alongside the CLIP loss.

---

### Official Review · Reviewer_v9wt · 2024-11-04

**Soundness:** 3
**Presentation:** 3
**Contribution:** 3
**Rating:** 6
**Confidence:** 3

**Summary:**

Brief Summary: The paper tackles the task of continual learning for a VLM. In particular, the authors introduce new datasets from image captioning datasets (coco, flickr30k) and the model is evaluated on a range of vision-language datasets on retrieval and zero-shot classification tasks.

The key idea here is that the authors show that LoRA (low rank adaptation) achieves a similar effect as storing and replaying old data. Further, that contrastive learning loss reduces forgetting.

Experiments on a number of datasets show that the proposed method (C-CLIP) trained can match or exceed full-finetuning results. Moreover, C-CLIP outperforms existing baselines by significant margins.

**Strengths:**

Pros:

1. The key idea of using LoRA as a replacement for memory bank and adding contrastive learning is easy to implement and is modular. As a result, it is promising that such method can be used to enable continual learning across a range of VLM models.

2. The authors contribute dataset and benchmarks. The performance improvement over competitive baselines is very significant (in range of 7-10 points).

3. The authors provide comprehensive set of experiments across multiple datasets and multiple baselines. The ablation study is also solid, particularly interesting is that Contrastive Loss (CKC) by itself provides significant improvement on its own (Table 4).

4. The authors provide qualitative results visualizing their model (in appendix)

**Weaknesses:**

Cons:

1. The authors should also show results with more multi-modal models and not just CLIP ViT-B/16, such as other CLIP variants with different backbones (say ViT-L/14).

2. The authors could also explore more vision-language tasks, in particular that of captioning/description with generative VLMs which output a description (such as LLaVA).

3.  (Minor) The authors have missed a close work [Ref1] which also explores the setting of multi-modal VLMs in continual learning setting. That said, the main contributions on modeling side are quite different.


=======

[Ref1]: Jin, Xisen, Junyi Du, Arka Sadhu, Ram Nevatia, and Xiang Ren. "Visually grounded continual learning of compositional phrases." arXiv preprint arXiv:2005.00785 (2020).

**Questions:**

Q1. It is a bit unclear to me if the model is provided the task id in the data-stream. For instance, during training does it know that the task is say pet-classification or is that information hidden.

---

> ### Author Response · Authors · 2024-11-24
> **Reply to Reviewer v9wt**
>
> We sincerely appreciate Reviewer v9wt for the review and are grateful for the time you spent with our submission. We are glad for the acknowledgment that our approach is promising, the proposed benchmarks are valuable and the experiments are solid. We wish to address your concerns by giving detailed responses to each of your comments as follows:
>
> **Q1.  Results with more multi-modal models, such as other CLIP variants with different backbones (say ViT-L/14).**
>
> Following your suggestions, we have implemented our method with different backbones, such as ViT-B/32, ViT-B/16, ViT-L/14 and ViT-L/14@336px, and the results are shown in Table 6 in the revised paper.  Specifically, due to memory constraints, we reduce the batch size to 256 for the larger ViT-L models. As shown in Table 6, ViT-L significantly outperforms ViT-B, with substantial improvements in zero-shot performance on downstream datasets. Our method performs well across various ViT architectures, maintaining strong zero-shot capabilities and excelling in image-text retrieval on downstream tasks.
>
>
> **Q2.  The authors could also explore more vision-language tasks, e.g., generative VLMs such as LLaVA.**
>
> Thanks for this suggestion. This paper focuses on classical version-language models like CLIP, which uses separate vision and text encoders aligned via contrastive loss to maximize the similarity between paired image-text embeddings. It is designed for tasks like retrieval and zero-shot classification.
>
> While generative VLMs like LLaVA integrate a vision module with large language models (LLM), in which the version module is frozen and only the LLM is fine-tuned on downstream datasets with language modeling autoregressive loss. Therefore, it is difficult to directly adapt the proposed constrastive knowledge consolidation to LLaVA. Nevertheless, considering the version encoder in LLaVA is from CLIP and plays an important role in LLaVA, we believe the proposed C-CLIP can also enhance the version ability of LLaVA.
>
> Inspired by your valuable suggestion, we added a discussion of this point in the revised paper, and will consider the continual learning for LLaVA in future work.
>
>
> **Q3.  (Minor) The authors have missed a close work [Ref1]. That said, the main contributions on modeling side are quite different.**
>
> Thanks for bringing this interesting work to our attention, and we have added it to the related work (section 2.1) of the revised version.
>
>
> **Q4. If the model is provided, is the task id in the data-stream? Does it know that the task is say pet-classification or is that information hidden?**
>
> The task id is not provided, and we do not need the task id information at inference. We focus on multimodal continual learning of VLMs and view each dataset as one task, similar to the domain-incremental learning in classical continual learning. From the perspective of methodology, on the one hand, we integrate the LoRA during the continual learning process, and we only have one set of LoRA after training. Therefore,  the task ID is not needed at inference time; on the other hand, the proposed CKC is based on constrastive loss, which also does not need the task ID information at both the training and inference stages. We have made this clear on Page 4 (Evaluation metric part) in the revised version.

---

### Meta-Review · Area_Chair_H8f6 · 2024-12-23

**Metareview:**

This paper introduces C-CLIP, a novel framework for continual learning in vision-language models, specifically addressing the challenge of maintaining zero-shot capabilities while learning new domains. The key claims include the effectiveness of combining LoRA adaptation with Contrastive Knowledge Consolidation (CKC), and the introduction of a new multimodal vision-language continual learning benchmark. The paper's strengths include addressing an important practical problem, strong empirical validation across multiple datasets, and comprehensive experiments showing significant improvements over baselines. The method demonstrates effective prevention of catastrophic forgetting while maintaining zero-shot capabilities. However, the paper had several initial weaknesses: unclear presentation of the benchmark definition, missing citations to relevant prior work, limited comparison with parameter-efficient methods, and some technical presentation issues. Based on the review scores (6, 6, 8, 8) and the extensive rebuttal responses, this paper warrants acceptance. The authors thoroughly addressed the concerns, added additional experiments with larger models, provided clearer benchmark definitions, and expanded comparisons with prompt-based methods

**Additional Comments On Reviewer Discussion:**

The reviewers raised several key points that were systematically addressed during rebuttal. Reviewer v9wt requested results with more model variants and clarification about task ID usage - the authors responded by adding experiments with ViT-B/32, ViT-L/14 and other architectures, and clarified that task IDs aren't needed at inference. Reviewer 8g5w questioned computational costs and comparisons with parameter-efficient methods - the authors added detailed computational analysis and new comparisons with L2P and CPE-CLIP methods. Reviewer L3rn provided extensive feedback on presentation and reproducibility concerns, which the authors addressed through significant revisions including new figures, clearer equations, and promises to release code. Reviewer 45wV raised concerns about missing citations and benchmark clarity - the authors added comprehensive citation coverage and reorganized the benchmark description with a new table. The authors' responses were thorough and substantive, adding significant new experimental results and clarifications that strengthened the paper.

---

### Decision · Program_Chairs · 2025-01-22

Accept (Poster)